



# Autonomous seawater $p$CO₂ and pH time series from 40 surface buoys and the emergence of anthropogenic trends

Adrienne J. Sutton[1], Richard A. Feely[1], Stacy Maenner-Jones[1], Sylvia Musielwicz[1,2], John Osborne[1,2], Colin Dietrich[1,2], Natalie Monacci[3], Jessica Cross[1], Randy Bott[1], Alex Kozyr[4], Andreas J. Andersson[5], Nicholas R. Bates[6,7], Wei-Jun Cai[8], Meghan F. Cronin[1], Eric H. De Carlo[9], Burke Hales[10], Stephan D. Howden[11], Charity M. Lee[12], Derek P. Manzello[13], Michael J. McPhaden[1], Melissa Meléndez[14,15], John B. Mickett[16], Jan A. Newton[16], Scott E. Noakes[17], Jae Hoon Noh[18], Solveig R. Olafsdottir[19], Joseph E. Salisbury[20], Uwe Send[5], Thomas W. Trull[21,22,23], Douglas C. Vandemark[20], Robert A. Weller[24]

[1]Pacific Marine Environmental Laboratory, National Oceanic and Atmospheric Administration, Seattle, Washington, USA
[2]Joint Institute for the Study of the Atmosphere and Ocean, University of Washington, Seattle, Washington, USA
[3]Ocean Acidification Research Center, University of Alaska Fairbanks, Fairbanks, Alaska, USA
[4]National Centers for Environmental Information, National Oceanic and Atmospheric Administration, Silver Spring, Maryland, USA
[5]Scripps Institution of Oceanography, University of California, San Diego, California, USA
[6]Bermuda Institute of Ocean Sciences, St. Georges, Bermuda
[7]Department of Ocean and Earth Science, University of Southampton, Southampton, UK
[8]University of Delaware, School of Marine Science and Policy, Newark, Delaware, USA
[9]University of Hawai'i at Mānoa, School of Ocean and Earth Science and Technology, Honolulu, Hawaii, USA
[10]College of Earth, Ocean and Atmospheric Sciences, Oregon State University, Corvallis, Oregon, USA
[11]Department of Marine Science, University of Southern Mississippi, Stennis Space Center, Mississippi, USA
[12]Ocean Policy Institute, Korea Institute of Ocean Science and Technology, Busan, Korea
[13]Atlantic Oceanographic and Meteorological Laboratory, National Oceanic and Atmospheric Administration, Miami, Florida, USA
[14]Department of Earth Sciences and Ocean Processes Analysis Laboratory, University of New Hampshire, Durham, New Hampshire, USA
[15]Caribbean Coastal Ocean Observing System, University of Puerto Rico, Mayagüez, Puerto Rico
[16]Applied Physics Laboratory, University of Washington, Seattle, Washington, USA
[17]Center for Applied Isotope Studies, University of Georgia, Athens, Georgia, USA
[18]Marine Ecosystem Research Center, Korea Institute of Ocean Science and Technology, Busan, Korea
[19]Marine and Freshwater Research Institute, Reykjavik, Iceland
[20]Ocean Process Analysis Laboratory, University of New Hampshire, Durham, New Hampshire, USA
[21]Climate Science Centre, Oceans and Atmosphere, Commonwealth Scientific and Industrial Research Organisation, Hobart, Australia
[22]Antarctic Climate and Ecosystems Cooperative Research Centre, Hobart, Australia
[23]Institute of Marine and Antarctic Studies, University of Tasmania, Hobart, Australia
[24]Woods Hole Oceanographic Institution, Woods Hole, Massachusetts, USA

*Correspondence to*: Adrienne J. Sutton (adrienne.sutton@noaa.gov)

**Abstract.** Ship-based time series, some now approaching over three decades long, are critical climate records that have dramatically improved our ability to characterize natural and anthropogenic drivers of ocean carbon dioxide (CO₂) uptake and biogeochemical processes. Advancements in autonomous marine carbon sensors and technologies over the last two decades have led to the expansion of observations at fixed time series sites, thereby improving the capability of characterizing sub-seasonal variability in the ocean. Here, we present a data product of 40 individual autonomous moored surface ocean $p$CO₂ (partial pressure of CO₂) time series established between 2004 and 2013, of which 17 also include autonomous pH measurements. These time series characterize a wide range of surface ocean carbonate conditions in different oceanic (17 sites), coastal (13 sites), and coral reef (10 sites) regimes. A time of trend emergence (ToE) methodology applied to the time series that exhibit well-constrained daily to interannual variability and an estimate of decadal variability indicates that the length of sustained observations necessary to detect statistically significant anthropogenic trends varies by marine environment. The ToE estimates for seawater $p$CO₂ and pH range



from 8 to 15 years at the open ocean sites, 16 to 41 years at the coastal sites, and 9 to 22 years at the coral reef sites. Only two open ocean $p$CO₂ time series, Woods Hole Oceanographic Institution Hawaii Ocean Time-series Station (WHOTS) in the subtropical North Pacific and Stratus in the South Pacific gyre, have been deployed longer than the estimated time of trend emergence and, for these, deseasoned monthly means show estimated anthropogenic trends of 1.9±0.3 µatm yr⁻¹ and 1.6±0.3 µatm yr⁻¹, respectively.

In the future, it is possible that updates to this product will allow for estimating anthropogenic trends at more sites; however, the product currently provides a valuable tool in an accessible format for evaluating climatology and natural variability of surface ocean carbonate chemistry in a variety of regions. Data are available at https://doi.org/10.7289/V5DB8043 and https://www.nodc.noaa.gov/ocads/oceans/Moorings/ndp097.html.

## 1 Introduction

Biogeochemical cycling leads to remarkable temporal and spatial variability of carbon in the mixed layer of the global ocean and particularly in coastal seas. The ocean carbon cycle, specifically surface ocean CO₂-carbonate chemistry, is primarily influenced by local physical conditions and biological processes, basin-wide circulation patterns, and fluxes between the ocean and land/atmosphere. Since the industrial period, increasing atmospheric CO₂ has been an additional forcing on ocean biogeochemistry, with the ocean absorbing roughly 30% of anthropogenic CO₂ (Khatiwala et al., 2013; Le Quéré et al., 2018). The resulting decrease

of seawater pH and carbonate ion concentration, referred to as ocean acidification, has the potential to impact marine life such as calcifying organisms (Bednaršek et al., 2017b; Chan and Connolly, 2013; Davis et al., 2017; Fabricius et al., 2011; Gattuso et al., 2015). Shellfish, shallow-water tropical corals, and calcareous plankton are a few examples of economically and ecologically important marine calcifiers potentially affected by ocean acidification.

Open ocean observations have shown that the inorganic carbon chemistry of the surface ocean is changing globally at a mean rate

consistent with atmospheric CO₂ increases of approximately 2.0 µatm yr⁻¹ (Bates et al., 2014; Takahashi et al., 2009; Wanninkhof et al., 2013). However, natural and anthropogenic processes can magnify temporal and spatial variability in some regions, especially coastal systems through eutrophication, freshwater input, exchange with tidal wetlands and the sea floor, seasonal biological productivity, and coastal upwelling (Bauer et al., 2013). This enhanced variability can complicate and at times obscure detection and attribution of longer-scale ocean carbon changes. There are also processes that can act in the opposite direction; for

example, riverine and estuarine sources of alkalinity increase buffering capacity of coastal waters and reduce the variability of other carbon parameters.

Efforts to observe and predict the impact of ocean acidification on marine ecosystems must be integrated with an understanding of both the natural and anthropogenic processes that control the ocean carbonate system. Marine organisms experience highly heterogeneous seawater carbonate chemistry conditions, and it is unclear what exact conditions in the natural environment will

lead to physiological responses (Hofmann et al., 2010). Although, responses associated with exposure to corrosive carbonate conditions such as low values of aragonite saturation state ($\Omega_{aragonite}$) have been observed (e.g., Barton et al., 2012, 2015; Bednaršek et al., 2014, 2016, 2017a; Reum et al., 2015). Observations show that present-day surface seawater pH and $\Omega_{aragonite}$ conditions throughout most of the open ocean exceed the natural range of preindustrial variability and in some coastal ecosystems, known biological thresholds for shellfish larvae are exceeded during certain times of the seasonal cycle (Sutton et al., 2016). Are these

present-day conditions significantly impacting marine life in the natural environment? How will intensity, frequency, and duration of corrosive carbonate conditions change as surface seawater pH and $\Omega_{aragonite}$ continue to decline and influence other processes of the biogeochemical cycle in the coastal zone? Paired chemical and biological observations at timescales relevant to biological





processes, such as food availability, seasonal spawning, larval growth, and recruitment, can be one tool for identifying and tracking the response of marine life to ocean acidification.

Long-term, sustained time-series observations resolving diurnal to seasonal conditions encompass many timescales relevant to biological processes and can help to characterize both natural variability and anthropogenic change in ocean carbon. Fixed time-
series observations fill a unique niche in ocean observing as they can serve as sites of multidisciplinary observations and process studies, high-quality reference stations for validating and assessing satellite measurements and earth system models, and test beds for developing and evaluating new ocean sensing technology. If of sufficient length and measurement quality to detect the anthropogenic signal above the noise (i.e., in this case the natural variability of the ocean carbon system), these observations can also serve as critical climate records.

Here, we introduce time-series data from 40 moored stations in open ocean, coastal, and coral reef environments. These time series include 3-hourly autonomous measurements of surface seawater temperature (SST), salinity (SSS), mole fraction of atmospheric $CO_2$ ($xCO_2$), partial pressure of atmospheric and seawater $CO_2$ ($pCO_2$), and seawater pH. This data product was developed to provide easy access to uninterrupted time series of high-quality $pCO_2$ and pH data for those who do not require the detailed deployment-level information archived at the National Centers for Environmental Information (NCEI;
https://www.nodc.noaa.gov/ocads/oceans/time_series_moorings.html).

We also present an overview of the seasonal variability to long-term trends revealed in the $pCO_2$ and pH observations, as well as an estimate of the length of time series required to detect an anthropogenic signal at each location. We use a statistical method described by Tiao et al. (1990) and further applied to environmental data by Weatherhead et al. (1998) to estimate the number of years of observations needed to detect a statistically significant trend over variability, which we refer to here as time of emergence
(ToE). An input required in this statistical model is an estimate of the trend. We adopt a trend in seawater $pCO_2$ of 2 $\mu$atm yr$^{-1}$, which assumes surface seawater changes track the current rate of globally-averaged atmospheric $CO_2$ increase. This assumption allows for comparing the trend-to-variance pattern across the network of 40 time series locations. The ToE methodology does not allow for identifying actual long-term trends that may be different from 2 $\mu$atm yr$^{-1}$ due to other long-term changes in, for example, biological production/respiration or coastal carbon sources/sinks. Nor does it address at what point in time a system may cross the
envelope of pre-industrial variability or biological thresholds (e.g., Pacella et al., 2018; Sutton et al, 2016). It indicates the time at which the imposed signal of 2 $\mu$atm yr$^{-1}$ emerges from the variance, and not necessarily when the actual anthropogenic signal may emerge or when organisms may be impacted.

Another caveat of this methodology is that the results apply to present-day conditions, and these estimates will change as the time
series lengthen due to continued anthropogenic forcing. For example, even if using seasonally detrended monthly anomalies (i.e., when the mean seasonality of ocean carbonate chemistry is accounted for), magnification of the seasonal amplitude of $pCO_2$ due to warming, reduction in buffering capacity, and/or other carbon cycle feedbacks could add variance to the monthly anomalies, resulting in increased detection time. Changes in circulation, stratification, and meltwater inputs in the Arctic cryosphere due to anthropogenic warming could also influence these estimates over time. For regions where the drivers of anthropogenic forcing
and natural variability are well constrained, the methodology could be modified to provide more accurate estimates of time of trend emergence. However, ToE estimates presented here use monthly anomalies of present-day observations and a fixed anthropogenic $pCO_2$ trend of 2 $\mu$atm yr$^{-1}$ to compare the trend-to-variance patterns across the network of 40 moored time series. These estimates provide a starting point for trend calculations using this data product.



## 2 Methods

### 2.1 Site and sensor description

The 40 fixed time series stations are located in the Pacific (29), Atlantic (9), Indian (1), and Southern (1) ocean basins in open ocean (17), coastal (13), and coral reef (10) ecosystems (Table 1; Fig. 1). All surface ocean $pCO_2$ and pH time series were

established between 2004 and 2013. Thirty-three of these stations are active, while three have been moved to nearby locations better representing regional biogeochemical processes and four have been discontinued due to lack of sustained funding. The range of support and partnerships for maintaining these moored time series is extensive; see Acknowledgements for details. Many of these 40 moored time series stations also make physical oceanographic and marine boundary layer meteorological measurements, thus enabling multi-disciplinary studies involving carbon cycle dynamics.

A Moored Autonomous $pCO_2$ (MAPCO$_2$) system measuring marine boundary layer air at 0.5–1 m height and seawater at <0.5 m depth is deployed at each fixed time series site (Sutton et al., 2014b). The MAPCO$_2$ systems measure $xCO_2$ in equilibrium with surface seawater by a nondispersive infrared gas analyzer (LI-COR: model LI-820) calibrated prior to each measurement with a reference gas traceable to World Meteorological Organization standards. Seawater $xCO_2$ equilibration occurs by cycling a closed loop of air through an equilibrator at the sea surface for 10 minutes. Each time series site has either a Sea-Bird Electronics (SBE)

16plus V2 Sea-CAT or a SBE 37 MicroCAT deployed at approximately 0.5 m measuring sea surface temperature (SST) and salinity (SSS). These measurements are used to calculate $pCO_2$ and the fugacity of $CO_2$ ($fCO_2$) consistent with standard operating procedures (Dickson et al., 2007; Weiss, 1974). Total estimated uncertainties of the resulting $pCO_2$ measurements are <2 µatm for seawater $pCO_2$ and <1 µatm for air $pCO_2$. For a detailed description of the MAPCO$_2$ methodology, calculations, data reduction, and data quality control, see Sutton et al. (2014b).

In addition to $pCO_2$, SST, and SSS, 17 of the time series also include seawater pH measurements at 0.5 m depth. These measurements are made by either the spectrophotometric-based Sunburst SAMI pH sensors (Seidel et al., 2008) or ion sensitive field effect transistor-based SeaFET pH sensors (Bresnahan et al., 2014; Martz et al., 2010). Field-based sensor validation suggests these sensors (once calibrated and adjusted in the case of the SeaFET) have a total uncertainty of <0.02 in this surface buoy application (Sutton et al., 2016). All seawater pH data are expressed in the total scale. At 3-hourly sampling intervals, this

configuration of MAPCO$_2$ and associated sensors is typically deployed for one year before recovery, maintenance, and redeployment of the buoy and sensors.

### 2.2 Data product description

All post-calibrated and quality-controlled data are archived at NCEI: https://www.nodc.noaa.gov/ocads/oceans/time_series_moorings.html. For each site, an annual deployment has data and quality

control descriptors at the data archive, including: (1) 3-hourly MAPCO$_2$ and associated data, including measured parameters such as $xCO_2$, humidity, and atmospheric pressure so data users can recalculate $pCO_2$ if desired; (2) a data quality flag (QF) log that identifies and describes likely bad (QF = 3) or bad (QF = 4) $CO_2$ and pH data included in the data set; and (3) a metadata file with deployment-level information such as reference gas value and MAPCO$_2$ air value comparisons to the GLOBALVIEW-CO$_2$ Marine Boundary Layer (MBL) product (GLOBALVIEW-CO$_2$, 2013). The reader is referred to Sutton et al. (2014b) for a detailed

description of this deployment-level archived information. In addition to data archival at NCEI, these deployment-level mooring data sets are also included in the annual Surface Ocean CO$_2$ Atlas data product (Bakker et al., 2016). Future data management plans include integrating the $pCO_2$ and pH data into OceanSITES, which would provide a single access point to open ocean



biogeochemical, physical oceanographic, and marine boundary layer meteorological measurements in a common, self-documented format.

The data product presented here is a compiled and simplified time series developed from these deployment-level archived files. Each fixed moored location has one file with a header including the following basic metadata: (1) data source and contact information; (2) data use request; (3) data product citation; (4) time series name, time range, and coordinates; (5) description of variables; (6) methodology references; and (7) links to deployment-level archived data and metadata at NCEI. Following the header, each fixed moored time series file includes the entire time series of SST, SSS, seawater $pCO_2$, air $pCO_2$, air $xCO_2$, and pH with an associated timestamp.

The time series data product only includes data from the original deployment-level data files assigned QF = 2 (good data). Any missing values or values assigned QF of 3 or 4 in the original deployment-level data are replaced with "NaN" in the time series product. Of the data assigned QF of 2, 3, or 4, the good data (QF = 2) retained in this data product comprise 96% of all seawater $xCO_2$ measurements and 88% of all seawater pH measurements. Missing or bad SST or SSS data further reduce the quantity of seawater $pCO_2$ values to 85% compared to the archived deployment-level data. Data users interested in all available $xCO_2$ and pH data should continue to retrieve deployment-level data from the NCEI archive.

Two time-series locations are exceptions to the above detail. Because 3-hourly SST and SSS are not available for the Twanoh and Dabob sites, the data archived at NCEI for those two sites includes $xCO_2$ (dry) air and seawater values but not calculated $pCO_2$. In order to calculate $pCO_2$ for those sites, the data user can incorporate atmospheric pressure, SST, and SSS from other sources. Atmospheric pressure at 3-hourly intervals can be found in the deployment-level archived data files at NCEI. Other data sources, including 2-hourly SST and SSS data at both Twanoh and Dabob, can also be located through the data portal of the Northwest Association of Networked Ocean Observing Systems: http://nvs.nanoos.org/. Since interpolating 2-hourly data with the 3-hourly MAPCO$_2$ data requires making assumptions about temporal variability that may differ according to the research interests of the data user, data from these two locations are only available in the deployment-level data files archived at NCEI.

This data product has been developed to provide easier access to quality-assured seawater $pCO_2$ and pH data and broaden the user base of these data. This data product is ideal for modelers interested in using fixed time series data to validate earth system model output or other data users accustomed to working with ship-based time series data. It also makes the time series more accessible to students, researchers from other disciplines, and marine resource managers who may not have a seawater $CO_2$-carbonate chemistry background or the resources necessary to process and interpret the more detailed deployment-level data.

### 2.3 Statistical analyses

Descriptive statistics from these time series products are presented here to compare variability in seawater $pCO_2$ and pH across the 40 locations. Seasonal amplitude is the difference in the mean of all observations during winter and summer. For Northern Hemisphere sites, winter is defined as December, January, and February, and summer is June, July, and August (vice versa for Southern Hemisphere sites).

The climatological mean is derived by averaging means for each of the 12 months over the composite, multiyear time series. Interannual variability (IAV) is presented as the standard deviation of individual yearly means throughout the time series. In the case of missing observations, climatological monthly means are substituted to calculate yearly means for IAV. This approach seeks to minimize the impact of data gaps on the IAV estimates. Because long-term trends in $pCO_2$ and pH are not well constrained at all locations, data are not detrended before calculating IAV. At Woods Hole Oceanographic Institution Hawaii Ocean Time-series



Station (WHOTS), for example, removing a trend of 2 μatm yr$^{-1}$ changes the IAV estimate by 12%. Therefore, IAV likely has high uncertainty due to lack of detrending, data gaps, and the relatively short time series lengths (≤12 years); future efforts will focus on improving these IAV estimates.

The seasonal cycle is removed from the data using the approaches described in detail in Bates (2001) and Takahashi et al. (2009). This method results in a time series of seasonally detrended monthly anomalies, which are monthly residuals after removing the climatological monthly means.

When applied to environmental data, ToE is a statistical method that estimates the number of years necessary in a time series to detect an anthropogenic signal over the natural variability. This method has been used to determine ToE from, for example, chlorophyll satellite records (Henson et al., 2010) and ocean biogeochemical models (Lovenduski et al., 2015). $ToE_{ts}$ (in years) of

each time series is derived using the method of Weatherhead et al. (1998):

$$ToE_{ts} = \left( \frac{3.3\sigma_N}{|\omega_0|} \sqrt{\frac{1+\varnothing}{1-\varnothing}} \right)^{2/3} \tag{1}$$

where $\sigma_N$ and $\varnothing$ are the standard deviation and autocorrelation (at lag 1) of monthly anomalies, respectively, and $\omega_0$ is the anthropogenic signal of 2 μatm $pCO_2$ or 0.002 pH per year, assuming surface seawater in equilibrium with the global mean rate of atmospheric $CO_2$ increase. This method results in a 90% probability (dictated by the factor of 3.3 in Eq. 1) of trend detection by

the estimated $ToE_{ts}$ at the 95% confidence interval. Uncertainty in $ToE_{ts}$, $u_{ToE}$, is calculated by:

$$u_{ToE} = ToE_{ts} \times e^B \tag{2}$$

where $B$ is the uncertainty factor calculated using the method of Weatherhead et al. (1998). Uncertainty is based on the number of months ($m$) in the time series and autocorrelation of monthly anomalies ($\varnothing$):

$$B = \frac{4}{3\sqrt{m}} \sqrt{\frac{1+\varnothing}{1-\varnothing}} \tag{3}$$

With time series lengths of ≤12 years, most of the moored time series characterize diurnal to interannual variability of surface ocean $pCO_2$; however, low-frequency decadal variability may not yet be fully captured. Decadal variability of surface ocean carbon is poorly quantified by observations in general (Keller et al., 2012; McKinley et al., 2011; Schuster and Watson, 2007; Séférian et al., 2013). In the absence of constraint of decadal variability at each of these locations, we consider an example in the tropical Pacific to estimate the impact of decadal variability on $ToE_{ts}$. For this example, we assume the decadal-scale forcing (i.e., primarily

the Pacific Decadal Oscillation; Newman et al., 2016) leads to a 27% change in $CO_2$ flux in the tropical Pacific (Feely et al. 2006). We take a conservative approach and assume this forcing is driven primarily by decadal changes in surface seawater $pCO_2$ of as much as 15% and determine the impact that added decadal variability has to the ToE estimates at the 7 sites on the Tropical Atmosphere Ocean (TAO) array (McPhaden et al, 1998). This is done by repeating the existing $pCO_2$ time series until time series length is 50 years and applying a 15% offset in the data on 10-year intervals at random. This simulated 50-year time series is then

used to recalculate ToE. The simulation with added low-frequency decadal signals increases ToE by an average of 40%, with significant variance across the TAO sites. Decadal forcing has less impact at the eastern Pacific TAO sites where subseasonal to interannual variability controlled by equatorial upwelling, tropical instability waves, and biological productivity is dominant, and more impact in the central and western Pacific where these higher-frequency modes of variability are less pronounced.

Decadal forcing may be particularly strong in the tropical Pacific due to the influence of the Pacific Decadal Oscillation on

equatorial upwelling of $CO_2$-rich water (Feely et al., 2006; Sutton et al., 2014a) compared to other subtropical sites (Keller et al.,



2012; Landschützer et al., 2016; Lovenduski et al., 2015; Schuster and Watson, 2007). However, we apply this 40% increase in $ToE_{ts}$ to all 40 time series in order to provide a conservative estimate of when an anthropogenic signal can be detected using these moored time series data. The reported ToE for each moored time series is the result from Eq. (1) multiplied by 1.4:

$$ToE = ToE_{ts} \times 1.4 \qquad\qquad (4)$$

For the data sets with time series length greater than these ToE estimates, monthly anomalies are linearly regressed against time to determine the long-term rate of change. Linear regression statistics, including uncertainty in rate and $r^2$, are calculated using standard methods described in Glover et al. (2011).

### 3 Results and Discussion

#### 3.1 Climatology and natural variability

Across the 40 moored stations, climatological means of surface ocean $pCO_2$ range from 255 to 490 µatm (Fig. 1). Seasonal amplitude of seawater $pCO_2$ vary from 8 to 337 µatm. With more recent establishment of seawater pH observations, only 10 of the 17 sites with pH sensors have seasonally-distributed pH data necessary to determine climatological mean and seasonal amplitude. At these 10 locations, climatological mean and seasonal amplitude of seawater pH vary from 8.00 to 8.21 and 0.01 to 0.14, respectively (Fig. 2). All the sites with seasonal amplitude reported in Figs. 1 and 2 have observations distributed across all seasons

(Fig. 3). Seasonal amplitude of surface seawater $pCO_2$ is largest at the coastal sites (60 to 337 µatm) compared to the open ocean (8 to 71 µatm) and coral reef sites (11 to 178 µatm). While seasonal pH variation is only constrained at 10 of the 40 sites, these patterns hold for pH as well with ranges of 0.08 to 0.14, 0.01 to 0.07, and 0.02 to 0.07 at the coastal, open ocean, and coral sites, respectively.

IAV of seawater $pCO_2$, which is the standard deviation of yearly means, range from 2 to 29 µatm. The largest IAV is found at the

coastal and coral sites with values at Coastal MS, Twanoh, and CRIMP2 of 29, 27, and 25 µatm, respectively. With a large IAV of 25 µatm, CRIMP2 tends to be an anomaly among coral sites, with most tropical coral locations exhibiting IAV similar to open ocean sites of ≤5 µatm (Fig. 1). Surface seawater pH time series are not yet long enough to determine a robust estimate of IAV.

These descriptive statistics show higher seawater $pCO_2$ values throughout the year in the tropical Pacific where equatorial upwelling of $CO_2$-rich water dominates. Seasonal forcing of $pCO_2$ values in this region is low, but IAV, driven by the El Niño

Southern Oscillation (Feely et al., 2006), is the highest of open ocean time series stations (Fig. 1). The coastal time series stations suggest annual $CO_2$ uptake with climatological means of seawater $pCO_2$ less than atmospheric $CO_2$ levels. Seasonal changes of SST and biological productivity drive the large seasonal amplitudes in $pCO_2$ and pH at the U.S. coastal locations (Fassbender et al., 2018; Reimer et al., 2017; Sutton et al., 2016; Xue et al., 2016). The coastal stations Twanoh and Coastal MS exhibit the highest IAV of seawater $pCO_2$ (reported as seawater $xCO_2$ for Twanoh) due to large variability from year to year in circulation, freshwater

input, and biological productivity (Fig. 1). Most coral reef time series stations suggest net annual calcification with positive $\Delta pCO_2$ (seawater – air) values. Net calcification has been confirmed by independent assessments at some of these coral reef time series stations (Bates et al., 2010; Courtney et al., 2016; Drupp et al., 2011; Shamberger et al., 2011).

Clusters of fixed time series stations in Washington and California State waters, the Hawaiian Island of Oahu, and Bermuda provide examples of how different processes drive ocean carbon chemistry. Seasonal amplitude and IAV are almost twice as large at the

time series stations within the freshwater-influenced Puget Sound (Dabob and Twanoh) compared to the stations on the outer coast of Washington (Chá bă and Cape Elizabeth; Fig. 1b). Dabob is closer to ocean source waters and is deeper compared to Twanoh,



which experiences greater water residence time and more persistent stratification, and therefore, increased influence of biological production and respiration on seawater $x\mathrm{CO_2}$ (Fassbender et al., 2018; Lindquist et al., 2017). These processes can cause subsurface hypoxia and low pH (<7.4) and aragonite saturation (<0.6) conditions in this region of Puget Sound (Feely et al., 2010), which likely contribute to the elevated surface seawater $x\mathrm{CO_2}$ levels observed at Dabob and Twanoh. The paired CCE1 and CCE2

moorings in coastal California provide the contrast of open ocean and upwelling regimes, respectively. Climatological mean and seasonal amplitude of $p\mathrm{CO_2}$ are higher at CCE2 where summer upwelling supplies $\mathrm{CO_2}$-rich water to the surface. IAV is similar at both sites, suggesting interannual drivers of $p\mathrm{CO_2}$, such as the El Niño Southern Oscillation (Nam et al., 2011), likely have an influence throughout the southern California Current Ecosystem.

In both Hawaii and Bermuda, coral reef time series stations are paired with offshore, open ocean $p\mathrm{CO_2}$ observatories, although the

offshore Bermuda Testbed Mooring (BTM) station was discontinued before the Bermuda reef sites were established. In both cases, the offshore stations of WHOTS and BTM both exhibit climatological mean seawater $p\mathrm{CO_2}$ slightly below atmospheric values (Fig. 1c), with previous studies indicating these locations are net annual $\mathrm{CO_2}$ sinks (Bates et al., 2014; Dore et al., 2003, 2009; Sutton et al., 2017). The fringing or outer reef sites in Oahu (Kilo Nalu, Ala Wai, Kaneohe) tend to exhibit seawater $p\mathrm{CO_2}$ values closer to these open ocean background levels. The lagoonal Oahu reefs (CRIMP1 and CRIMP2) reflect increased water retention

time paired with coral reef photosynthesis/respiration and calcification/dissolution, which elevate both annual mean and daily to interannual variability in seawater $p\mathrm{CO_2}$ values (Fig. 1c; Courtney et al., 2017; Drupp et al., 2011, 2013). One exception is the nearly as large IAV at the fringing reef Ala Wai site, which is impacted by a nearby urban canal/estuary with high nutrient and organic matter input during storm events (Drupp et al., 2013). Positive $\Delta p\mathrm{CO_2}$ values at the lagoonal reef sites also suggest that these sites are a net source of $\mathrm{CO_2}$ to the atmosphere in contrast to the annual net $\mathrm{CO_2}$ uptake at the nearby open ocean sites (Fig.

1c).

In contrast, the outer reef site in Bermuda (Hog Reef) has a higher seasonal amplitude and mean $p\mathrm{CO_2}$ than the inner reef (Crescent Reef) despite having a shorter water residence time (Fig. 1). This is due to the greater biomass at Hog Reef, reflecting the influence of short-term (~1-2 days) of the local active reef community, whereas Crescent Reef reflects the integrated signal of multiple habitats and days (~6 days; Takeshita et al., 2018). Another caveat is the coral reef time series in this data product have an inherent

spatial bias as 80% of the coral reef moorings are located > 20° latitude. The patterns for cooler, high-latitude reefs (e.g., Oahu and Bermuda) may differ from lower latitude reef sites (e.g., La Parguera and Chuuk), which would generally have less pronounced seasonality.

### 3.2 Marine boundary layer atmospheric $CO_2$

Atmospheric $\mathrm{CO_2}$ observations at the 40 time series sites all show a positive long-term trend (Fig. 4a). The mean trend at the open

ocean sites are not significantly different from the global average rate of change of 2 ppm yr$^{-1}$ (Sutton et al., 2014b). Fig. 4a shows all 40 time series of atmospheric $x\mathrm{CO_2}$ with a rate of change of approximately 20 µmol mol$^{-1}$ (or ppm) over a decade; that is, from 380 µmol mol$^{-1}$ in January 2006 to 400 µmol mol$^{-1}$ in January 2016.

Although the global observing network of atmospheric $\mathrm{CO_2}$ that tracks anthropogenic $\mathrm{CO_2}$ increase requires higher measurement quality ($\leq 0.1$ ppm) compared to the measurement quality of the MAPCO$_2$ system ($\leq 1$ ppm), the MAPCO$_2$ air data may be valuable

for regional air $\mathrm{CO_2}$ studies in coastal regions where land-based activities cause larger hourly to interannual variability in atmospheric $\mathrm{CO_2}$ (Bender et al., 2002). In general, the coastal stations exhibit higher annual mean and seasonal amplitude compared to GLOBALVIEW-$\mathrm{CO_2}$ MBL values, which is a product based on interpolating high-quality atmospheric measurements around the globe to latitudinal distributions of biweekly $\mathrm{CO_2}$ (Fig. 4b,c). Open ocean and coral reef sites do not show this overall pattern



compared to GLOBALVIEW-$CO_2$ MBL values, although there is variability across the sites with some time series exhibiting higher means and seasonal amplitudes compared to the data product and vice versa (Fig. 4b,c).

### 3.3 Emergence of anthropogenic trends in surface seawater $p\mathrm{CO_2}$ and pH

Estimated length of time for an anthropogenic trend in seawater $p\mathrm{CO_2}$ to emerge from natural variability in the 40 time series varies
from 8 to 41 years (Fig. 5). This range is 8 to 15 years at the open ocean sites, 16 to 41 years at the coastal sites, and 9 to 22 years at the coral reef sites. For the pH data sets with long enough time series to calculate ToE (i.e., the circles in Fig. 2), there is no significant difference between ToE of $p\mathrm{CO_2}$ and pH (ToE calculated using hydrogen ion concentration, $[\mathrm{H^+}]$, not $-\log[\mathrm{H^+}]$), therefore, it is likely that ToE presented in Fig. 5 signifies both surface seawater $p\mathrm{CO_2}$ and pH. However, as the pH time series lengthen and variability is better constrained, future work should focus on a more thorough assessment of ToE of seawater pH.

Since ToE is dependent on the variability in the data distinct from the secular anthropogenic trend, the sites that exhibit higher variability (Figs. 1 and 2) tend to have longer ToE estimates (Fig. 5). The fringing and outer reef sites of south shore Oahu (Kilo Nalu and Ala Wai) and Kaneohe Bay, respectively, have shorter ToE compared to the lagoonal sites (CRIMP1 and CRIMP2) with larger seasonal to interannual variability. Similarly, the freshwater-influenced, highly-productive Puget Sound sites (Dabob and Twanoh) have the longest ToE of all 40 sites and are approximately twice as long as the nearby time series on the outer coast of
Washington (Chá bă and Cape Elizabeth). In the southern California Current, the ToE of the upwelling-influenced CCE2 is 50% longer than the offshore CCE1 site.

These data also suggest that removing seasonal variability from the times series is essential to reducing ToE and determining accurate long-term trends. The ToE estimates presented in Fig. 5 are based on seasonally detrended monthly anomalies, which are the residuals of the climatological monthly means. These ToE estimates are on average 55% shorter than ToE estimated using raw
time series data. This reduction in ToE due to seasonally detrending has a larger impact at higher latitudes where the seasonal amplitude of surface seawater $p\mathrm{CO_2}$ is larger compared to tropical sites. Using anomalies of climatological monthly means also minimizes the impact of start and end month of the time series on the resulting trend estimation.

Of the 40 seawater $p\mathrm{CO_2}$ time series, ToE estimates suggest only the WHOTS and Stratus time series are currently long enough to detect an anthropogenic trend. KEO, Papa, Kilo Nalu, and some TAO time series are approaching ToE, but at this time final data
are not yet available through 2017. Data available at the time of publication suggest the anthropogenic trend in surface seawater $p\mathrm{CO_2}$ at WHOTS from 2004 to 2014 is $1.9\pm0.3$ μatm yr$^{-1}$ (Fig. 6). In this trend analysis we do not include data from the 2014-2015 anomalous event that warmed North Pacific Ocean surface water (Bond et al., 2015) and elevated seawater $p\mathrm{CO_2}$ values (Feely et al., 2017). This WHOTS trend is not significantly different from the seawater $p\mathrm{CO_2}$ trend observed from 1988 to 2013 at the collocated ship-based Station ALOHA of $2.0\pm0.1$ μatm yr$^{-1}$ (Sutton et al., 2017). Both WHOTS and Station ALOHA trends are
not significantly different from the trend expected if surface seawater is in equilibrium with the global average atmospheric $CO_2$ increase.

The long-term trend at Stratus from 2006 to 2015 is $1.6\pm0.3$ μatm yr$^{-1}$ (Fig. 6). This trend is slightly lower than expected if seawater $p\mathrm{CO_2}$ change is in equilibrium with the atmosphere. Considering the uncertainty in the ToE$_{ts}$ estimate (Table 2) and the added uncertainty around unconstrained decadal variability at each of these locations, continued observations will be necessary at this
site to confirm whether this lower rate of change persists. In addition to uptake of atmospheric $CO_2$, the seawater $p\mathrm{CO_2}$ trend may be impacted by surface meteorological or upper ocean changes in this region. Significant trends in wind speed, wind stress, and the air–sea exchange of heat, freshwater, and momentum were observed from meteorological and surface ocean measurements on Stratus from 2000 to 2009 (Weller, 2015). These trends are related to intensification of Pacific trade winds over the last two decades



across the entire basin (England et al., 2014) and are likely to impact surface ocean $p$CO$_2$ and CO$_2$ flux in other regions of the Pacific. Sustained, continuous time series such as Stratus can contribute to constraining the physical and biogeochemical processes controlling long-term change.

## 4 Data Availability

Locations of deployment-level archived data at NCEI and the time series data product for each mooring site are listed in Table 2. The Digital Object Identifier (DOI) for this data product is:10.7289/V5DB8043. Data users looking for easier access to quality-assured seawater $p$CO$_2$ and pH data designated good (QF = 2; see Sect. 2.2) should consider using this time series data product. The time series data files will be updated each time new deployment-level data are submitted to the NCEI archive. Data users interested in all available MAPCO$_2$ and pH data should retrieve deployment-level data at NCEI (links also provided in Table 2).

These data are made freely available to the public and the scientific community in the belief that their wide dissemination will lead to greater understanding and new scientific insights. Users of these time series data products should reference this paper and acknowledge the major funding organizations of this work: NOAA's Ocean Observing and Monitoring Division and Ocean Acidification Program.

## 5 Conclusions

This product provides a unique data set for a range of users including providing a more accessible format for non-carbon chemists interested in surface ocean $p$CO$_2$ and pH time series data. These 40 time series locations represent a range of ocean, coastal, and coral reef regimes that exhibit a broad spectrum of daily to interannual variability. These time series can be used as a tool for estimating climatologies, assessing natural variability, and constraining models to improve predictions of trends in these regions. However, at this time, only two time series data sets (WHOTS and Stratus) are long enough to estimate long-term anthropogenic

trends. ToE estimates show at all but these two sites, an anthropogenic signal cannot be discerned at a statistically significant level from the natural variability of surface seawater $p$CO$_2$ and pH. If and when that date of trend emergence is attained, it is essential to seasonally detrend data prior to any trend analyses. Even though the ToE provided are conservative estimates, data users should still use caution in interpreting that an anthropogenic trend is distinct from decadal-scale ocean forcing that is not well characterized. Future work should be directed at improving upon these ToE estimates in regions where other data, proxies, or

knowledge about decadal forcing are more complete.

## Acknowledgements

We gratefully acknowledge the major funders of the $p$CO$_2$ and pH observations: the Office of Oceanic and Atmospheric Research of the National Oceanic and Atmospheric Administration, U.S. Department of Commerce, including resources from the Ocean Observing and Monitoring Division of the Climate Program Office (fund reference number 100007298) and the Ocean

Acidification Program. We rely on a long list of scientific partners and technical staff who carry out buoy maintenance, sensor deployment, and ancillary measurements at sea. We thank these partners and their funders for their continued efforts in sustaining the platforms that support these long-term $p$CO$_2$ and pH observations, including: Australian Integrated Marine Observing System, Caribbean Coastal Ocean Observing System, Gray's Reef National Marine Sanctuary, Marine and Freshwater Research Institute, Murdock Charitable Trust, National Data Buoy Center, National Science Foundation Division of Ocean Sciences, NOAA–Korean

Ministry of Oceans and Fisheries Joint Project Agreement, Northwest Association of Networked Ocean Observing Systems,




Research Moored Array for African-Asian-Australian Monsoon Analysis and Prediction (i.e., RAMA), University of Washington, U.S. Integrated Ocean Observing System, and the Washington Ocean Acidification Center. The open ocean sites are part of the OceanSITES program of the Global Ocean Observing System and the Surface Ocean $CO_2$ Observing Network. All sites are also part of the Global Ocean Acidification Observing Network. This paper is PMEL contribution number 4797.

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



**Table 1: Region, coordinates, surface ocean carbon parameters measured, year carbon time series established, and current status of the 40 fixed moored time series stations. All time series also include atmospheric CO₂, SST, and SSS.**

| Abbreviation | Descriptive name | Region | Latitude | Longitude | Carbon parameters | Start year | Status |
|---|---|---|---|---|---|---|---|
| CCE1 | California Current Ecosystem 1 | Northeast Pacific Ocean | 33.48 | -122.51 | $p$CO₂, pH | 2008 | active |
| Papa | Ocean Station Papa | Northeast Pacific Ocean | 50.13 | -144.84 | $p$CO₂, pH | 2007 | active |
| KEO | Kuroshio Extension Observatory | Northwest Pacific Ocean | 32.28 | 144.58 | $p$CO₂, pH | 2007 | active |
| JKEO | Japanese Kuroshio Extension Observatory | Northwest Pacific Ocean | 37.93 | 146.52 | $p$CO₂ | 2007 | discontinued in 2007 |
| WHOTS | Woods Hole Oceanographic Institution Hawaii Ocean Time-series Station | Central Pacific Ocean | 22.67 | -157.98 | $p$CO₂, pH | 2004 [a] | active |
| TAO110W | National Data Buoy Center (NDBC) Tropical Atmosphere Ocean 0º, 110ºW | Equatorial Pacific Ocean | 0.00 | -110.00 | $p$CO₂ | 2009 | active |
| TAO125W | NDBC Tropical Atmosphere Ocean 0º, 125ºW | Equatorial Pacific Ocean | 0.00 | -125.00 | $p$CO₂ | 2004 | active |
| TAO140W | NDBC Tropical Atmosphere Ocean 0º, 140ºW | Equatorial Pacific Ocean | 0.00 | -140.00 | $p$CO₂ | 2004 | active |
| TAO155W | NDBC Tropical Atmosphere Ocean 0º, 155ºW | Equatorial Pacific Ocean | 0.00 | -155.00 | $p$CO₂ | 2010 | active |
| TAO170W | NDBC Tropical Atmosphere Ocean 0º, 170ºW | Equatorial Pacific Ocean | 0.00 | -170.00 | $p$CO₂ | 2005 | active |
| TAO165E | NDBC Tropical Atmosphere Ocean 0º, 165ºE | Equatorial Pacific Ocean | 0.00 | 165.00 | $p$CO₂ | 2010 | active |
| TAO8S165E | NDBC Tropical Atmosphere Ocean 8ºS, 165ºE | Equatorial Pacific Ocean | -8.00 | 165.00 | $p$CO₂ | 2009 | active |
| Stratus | Stratus | Southeast Pacific Ocean | -19.70 | -85.60 | $p$CO₂, pH | 2006 | active |
| BTM | Bermuda Testbed Mooring | North Atlantic Ocean | 31.50 | -64.20 | $p$CO₂ | 2005 | discontinued in 2007 |
| Iceland | North Atlantic Ocean Acidification Mooring | North Atlantic Ocean | 68.00 | -12.67 | $p$CO₂, pH | 2013 | active |
| BOBOA | Bay of Bengal Ocean Acidification Observatory | Indian Ocean | 15.00 | 90.00 | $p$CO₂, pH | 2013 | active |
| SOFS | Southern Ocean Flux Station | Southern Ocean | -46.80 | 142.00 | $p$CO₂ | 2011 | active |
| GAKOA | Gulf of Alaska Ocean Acidification Mooring | Alaskan Coast | 59.910 | -149.350 | $p$CO₂, pH [b] | 2011 | active |
| Kodiak | Kodiak Alaska Ocean Acidification Mooring | Alaskan Coast | 57.700 | -152.310 | $p$CO₂, pH [b] | 2013 | discontinued in 2016 |
| SEAK | Southeast Alaska Ocean Acidification Mooring | Alaskan Coast | 56.260 | -134.670 | $p$CO₂, pH [b] | 2013 | discontinued in 2016 |
| M2 | Southeastern Bering Sea Mooring Site 2 | Bering Sea Coastal Shelf | 56.510 | -164.040 | $p$CO₂, pH [b] | 2013 | active |
| Cape Elizabeth | NDBC Buoy 46041 in Olympic Coast National Marine Sanctuary (NMS) | U.S. West Coast | 47.353 | -124.731 | $p$CO₂ | 2006 | active |
| Chá bǎ | Chá bǎ Buoy in the Northwest Enhanced Moored Observatory and Olympic Coast NMS | U.S. West Coast | 47.936 | -125.958 | $p$CO₂, pH | 2010 | active |
| CCE2 | California Current Ecosystem 2 | U.S. West Coast | 34.324 | -120.814 | $p$CO₂, pH | 2010 | active |
| Dabob | Oceanic Remote Chemical Analyzer (ORCA) buoy at Dabob in Hood Canal | U.S. West Coast | 47.803 | -122.803 | $x$CO₂ [c] | 2011 | active |
| NH-10 | Newport Hydrographic Line Station 10 Ocean Acidification Mooring | U.S. West Coast | 44.904 | -124.778 | $p$CO₂, pH | 2014 | moved to new location in 2017 [d] |
| Twanoh | ORCA buoy at Twanoh in Hood Canal | U.S. West Coast | 47.375 | -123.008 | $x$CO₂ [c] | 2009 | active |
| Ala Wai | Ala Wai Water Quality Buoy at South Shore Oahu | Pacific Island Coral Reef | 21.280 | -157.850 | $p$CO₂ | 2008 | active |
| Chuuk | Chuuk Lagoon Ocean Acidification Mooring | Pacific Island Coral Reef | 7.460 | 151.900 | $p$CO₂, pH | 2011 | active |
| CRIMP1 | Coral Reef Instrumented Monitoring Platform 1 | Pacific Island Coral Reef | 21.428 | -157.788 | $p$CO₂ | 2005 | moved to CRIMP2 in 2008 |
| CRIMP2 | Coral Reef Instrumented Monitoring Platform 2 | Pacific Island Coral Reef | 21.458 | -157.798 | $p$CO₂ | 2008 | active |
| Kaneohe | Kaneohe Bay Ocean Acidification Offshore Observatory | Pacific Island Coral Reef | 21.480 | -157.780 | $p$CO₂, pH | 2011 | active |
| Kilo Nalu | Kilo Nalu Water Quality Buoy at South Shore Oahu | Pacific Island Coral Reef | 21.288 | -157.865 | $p$CO₂ | 2008 | active |
| Gray's Reef | NDBC Buoy 41008 in Gray's Reef National Marine Sanctuary | U.S. East Coast | 31.400 | -80.870 | $p$CO₂, pH | 2006 | active |
| Gulf of Maine | Coastal Western Gulf of Maine Mooring | U.S. East Coast | 43.023 | -70.542 | $p$CO₂, pH | 2006 | active |





| Crescent Reef | Crescent Reef Bermuda Buoy | Atlantic Coral Reef | 32.400 | -64.790 | $p$CO$_2$ | 2010 | active |
|---|---|---|---|---|---|---|---|
| Hog Reef | Hog Reef Bermuda Buoy | Atlantic Coral Reef | 32.460 | -64.830 | $p$CO$_2$ | 2010 | active |
| Coastal MS | Central Gulf of Mexico Ocean Observing System Station 01 | Gulf of Mexico Coast | 30.000 | -88.600 | $p$CO$_2$, pH | 2009 | moved to new location in 2017 [e] |
| Cheeca Rocks | Cheeca Rocks Ocean Acidification Mooring in Florida Keys National Marine Sanctuary | Caribbean Coral Reef | 24.910 | -80.624 | $p$CO$_2$, pH | 2011 | active |
| La Parguera | La Parguera Ocean Acidification Mooring | Caribbean Coral Reef | 17.954 | -67.051 | $p$CO$_2$, pH | 2009 | active |

Notes: [a] Data from December 2004 to July 2007 in the WHOTS time series are from the Multi-disciplinary Ocean Sensors for Environmental Analyses and Networks (MOSEAN) station at 22.80ºN, 158.10ºW (20 km from the WHOTS location). Previous studies have shown the MOSEAN and WHOTS locations have similar surface seawater $p$CO$_2$ conditions (Sutton et al., 2014b, 2017) and are therefore combined in this data product as one time series location. [b] Measurements of pH to be included in future updates of the time series data product. [c] SST and SSS data are collected on the Dabob and Twanoh buoys at 2-hourly intervals. Because combining these data with the 3-hourly MAPCO$_2$ data requires making assumptions about temporal variability that reflect the research interests of the data user, only the direct measurements of CO$_2$ (i.e., the mole fraction of CO$_2$ in equilibrium with surface seawater: $x$CO$_2$) are available in the NCEI archived data sets. [d] The NH-10 buoy and carbon sensors were moved approximately 75 nm south to Cape Arago, Oregon, following establishment of an Ocean Observatories Initiative buoy at NH-10 with redundant $p$CO$_2$ and pH sensors: https://www.pmel.noaa.gov/co2/story/CB-06. [e] The Coastal MS buoy and carbon sensors were moved approximately 115 nm southwest to coastal Louisiana waters: https://www.pmel.noaa.gov/co2/story/Coastal+LA.



**Table 2: Data access to deployment-level archived data files at NCEI and the time series data product for each moored buoy location. The earliest date of seawater $p$CO$_2$ trend emergence is based on time series product data and calculated by adding the ToE estimate (Eqs. 1-4) to the time series start year (Table 1). The uncertainty presented here is the result of Eqs. (2-3), which is based on ToE$_{ts}$ and does not include any additional uncertainty due to the decadal estimate from Eq. (4). NA denotes sites with less than 3 years of observations where interannual variability is likely not represented in a time series, and therefore, ToE is not calculated.**

| Buoy name | NCEI archived data files (https://www.nodc.noaa.gov/…) | Time series data product (https://www.pmel.noaa.gov/co2/…) | Earliest date of seawater $p$CO$_2$ trend emergence |
|---|---|---|---|
| CCE1 | ocads/data/0144245.xml | timeseries/CCE1.txt | 2020 ± 2 |
| Papa | ocads/data/0100074.xml | timeseries/PAPA.txt | 2017 ± 2 |
| KEO | ocads/data/0100071.xml | timeseries/KEO.txt | 2018 ± 2 |
| JKEO | ocads/data/0100070.xml | timeseries/JKEO.txt | NA[a] |
| WHOTS | ocads/data/0100073.xml[b] ocads/data/0100080.xml | timeseries/WHOTS.txt | 2013 ± 1 |
| TAO110W | ocads/data/0112885.xml | timeseries/TAO110W.txt | 2024 ± 4 |
| TAO125W | ocads/data/0100076.xml | timeseries/TAO125W.txt | 2017 ± 4 |
| TAO140W | ocads/data/0100077.xml | timeseries/TAO140W.txt | 2018 ± 2 |
| TAO155W | ocads/data/0100084.xml | timeseries/TAO155W.txt | NA |
| TAO170W | ocads/data/0100078.xml | timeseries/TAO170W.txt | 2016 ± 4 |
| TAO165E | ocads/data/0113238.xml | timeseries/TAO165E.txt | NA |
| TAO8S165E | ocads/data/0117073.xml | timeseries/TAO8S165E.txt | 2021 ± 2 |
| Stratus | ocads/data/0100075.xml | timeseries/STRATUS.txt | 2015 ± 1 |
| BTM | ocads/data/0100065.xml | timeseries/BTM.txt | NA[a] |
| Iceland | ocads/data/0157396.xml | timeseries/ICELAND.txt | NA |
| BOBOA | ocads/data/0162473.xml | timeseries/BOBOA.txt | NA |
| SOFS | ocads/data/0118546.xml | timeseries/SOFS.txt | NA |
| GAKOA | ocads/data/0116714.xml | timeseries/GAKOA.txt | 2027 ± 3 |
| Kodiak | ocads/data/0157347.xml | timeseries/KODIAK.txt | 2028 ± 3[a] |
| SEAK | ocads/data/0157601.xml | timeseries/SEAK.txt | NA[a] |
| M2 | ocads/data/0157599.xml | timeseries/M2.txt | NA |
| Cape Elizabeth | ocads/data/0115322.xml | timeseries/CAPEELIZABETH.txt | 2030 ± 4 |
| Chá bă | ocads/data/0100072.xml | timeseries/CHABA.txt | 2033 ± 4 |
| CCE2 | ocads/data/0084099.xml | timeseries/CCE2.txt | 2028 ± 3 |
| Dabob | ocads/data/0116715.xml | *use NCEI files* | 2050 ± 6 |
| NH-10 | ocads/data/0157247.xml | timeseries/NH10.txt | NA[a] |
| Twanoh | ocads/data/0157600.xml | *use NCEI files* | 2050 ± 6 |
| Ala Wai | ocads/data/0157360.xml | timeseries/ALAWAI.txt | 2024 ± 3 |
| Chuuk | ocads/data/0157443.xml | timeseries/CHUUK.txt | 2021 ± 2 |



| | | | |
|---|---|---|---|
| CRIMP1 | ocads/data/0100069.xml | timeseries/CRIMP1.txt | $2022 \pm 4^{a}$ |
| CRIMP2 | ocads/data/0157415.xml | timeseries/CRIMP2.txt | $2030 \pm 3$ |
| Kaneohe | ocads/data/0157297.xml | timeseries/KANEOHE.txt | *NA* |
| Kilo Nalu | ocads/data/0157251.xml | timeseries/KILONALU.txt | $2017 \pm 2$ |
| Gray's Reef | ocads/data/0109904.xml | timeseries/GRAYSREEF.txt | $2027 \pm 3$ |
| Gulf of Maine | ocads/data/0115402.xml | timeseries/GULFOFMAINE.txt | $2023 \pm 3$ |
| Crescent Reef | ocads/data/0117059.xml | timeseries/CRESCENTREEF.txt | $2020 \pm 2$ |
| Hog Reef | ocads/data/0117060.xml | timeseries/HOGREEF.txt | $2023 \pm 3$ |
| Coastal MS | ocads/data/0100068.xml | timeseries/COASTALMS.txt | $2046 \pm 7$ |
| Cheeca Rocks | ocads/data/0157417.xml | timeseries/CHEECAROCKS.txt | $2020 \pm 2$ |
| La Parguera | ocads/data/0117354.xml | timeseries/LAPARGUERA.txt | $2019 \pm 2$ |

Notes: a  Discontinued sites where a long-term trend cannot be quantified solely from this time series data product. b  Links to NCEI archived deployment-level data files are provided for both MOSEAN and WHOTS; however, these time series are combined in the time series data product.



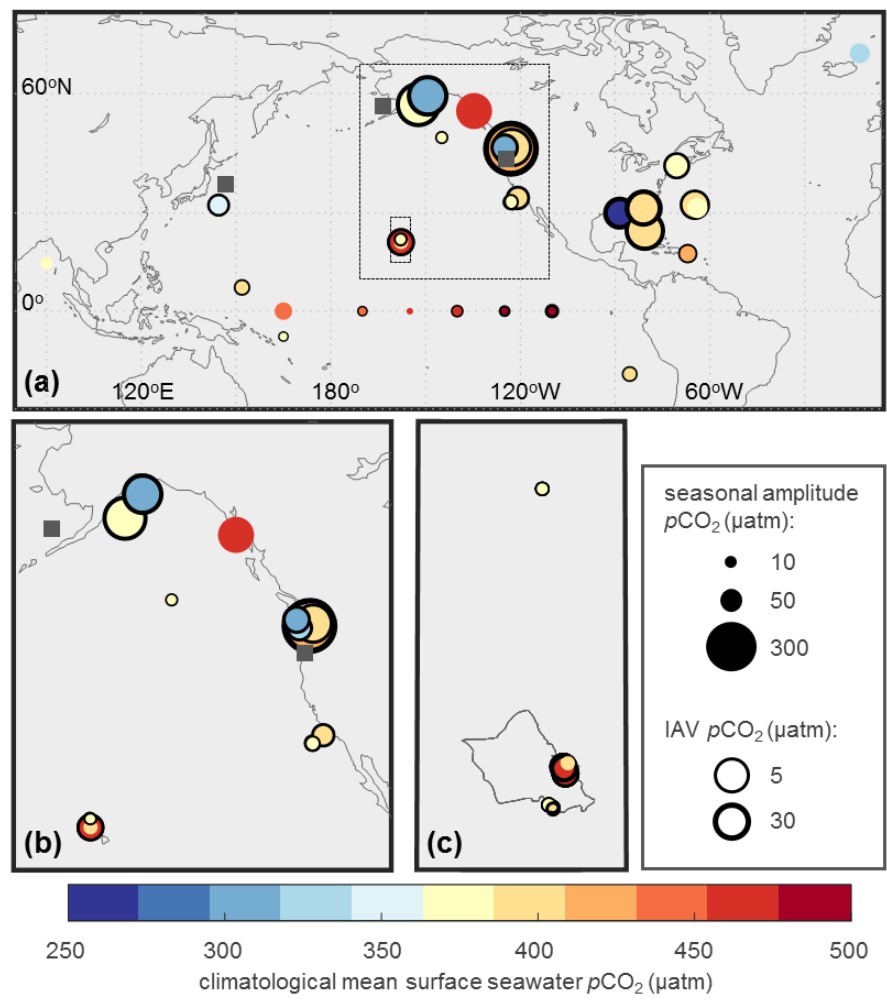

**Figure 1:** Location of (a) 40 moored $p$CO$_2$ time series with insets enlarged for the (b) U.S. West Coast and (c) Hawaiian Island of Oahu.
Circle color represents climatological mean seawater $p$CO$_2$ (µatm), size of circle represents seasonal amplitude, and thickness of circle
outline represents interannual variability (IAV). Gray squares show the locations of JKEO, M2, and NH-10 where insufficient winter
observations prevent the calculation of climatological mean or seasonal amplitude. IAV is not shown for sites with less than 3 years of
observations (Kaneohe, Iceland, BOBOA, SEAK, M2, SOFS, BTM, TAO165E, TAO155W, NH-10, and JKEO). Dabob and Twanoh data
shown here are $x$CO$_2$ (µmol mol$^{-1}$).





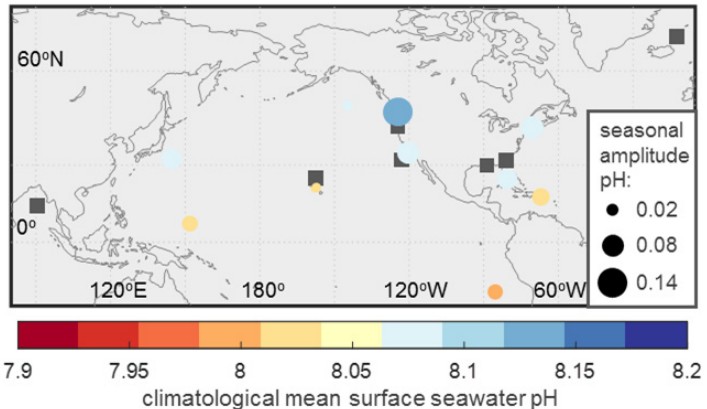

**Figure 2: Location of 17 moored pH time series. Circle color represents climatological mean seawater pH and size of circle represents seasonal amplitude. Gray squares show the locations of moored pH time series where lack of seasonal distribution of measurements prevent the calculation of climatological mean or seasonal amplitude. No pH time series are of sufficient length to estimate IAV as presented for seawater $p$CO₂ in Figure 1.**





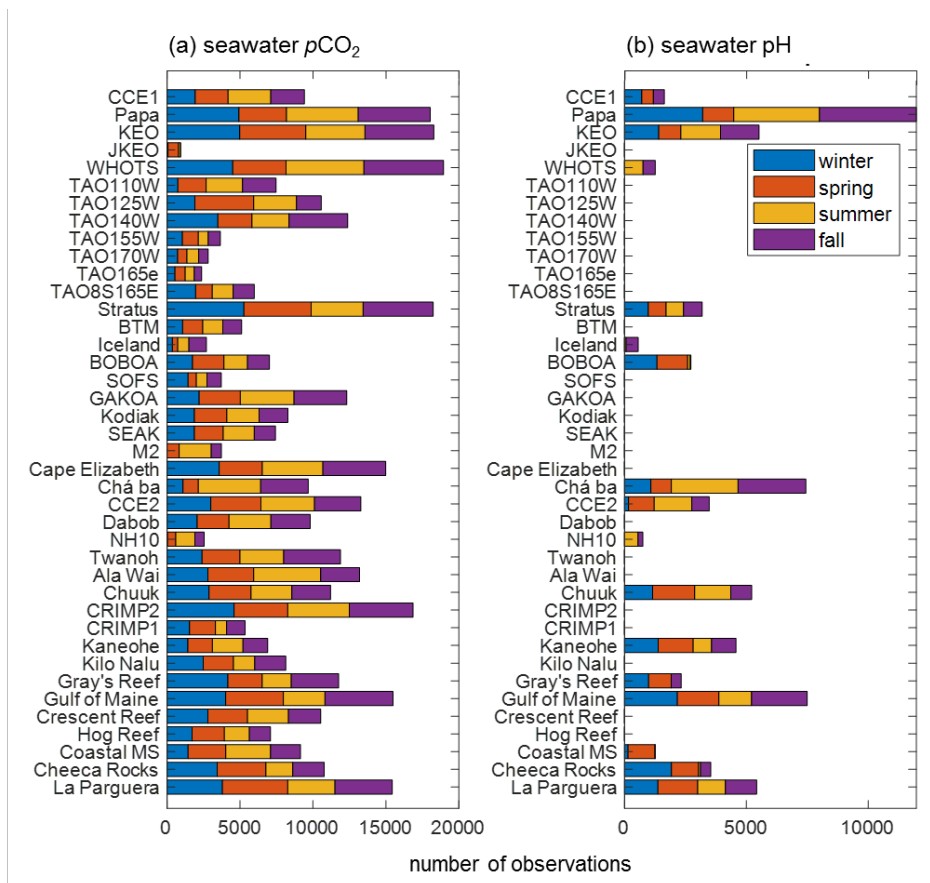

Figure 3: Number of surface seawater (a) $pCO_2$ and (b) pH observations by season in each of the 40 moored time series. For Northern Hemisphere sites, winter is defined as December, January, February; spring is March, April, May; summer is June, July, August; and fall is September, October, November (seasons reversed for Southern Hemisphere sites). Number of observations for Dabob and Twanoh shown here are seawater $xCO_2$.





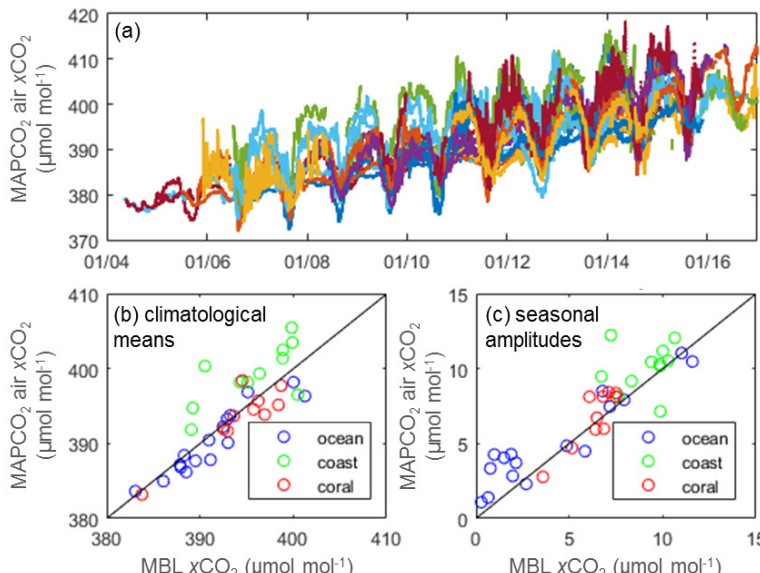

**Figure 4: (a) Weekly averaged air $x$CO$_2$ observations from the 40 time series. Different colors represent different time series. Dates are MM/YY. (b) Climatological means and (c) seasonal amplitudes of air $x$CO$_2$ from the MAPCO$_2$ measurements compared to the GLOBALVIEW-CO2 MBL data product (GLOBALVIEW-CO2, 2013) for open ocean (blue), coastal (green), and coral reef (red) time series locations.**



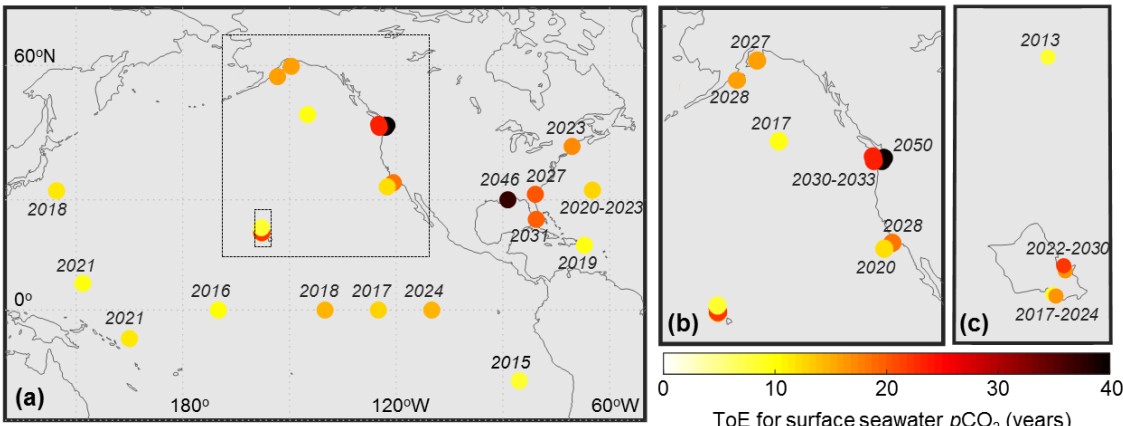

**Figure 5: (a) Time of trend emergence (ToE) estimates in years with insets enlarged for the (b) U.S. West Coast and (c) Hawaiian Island of Oahu. ToE is not shown for sites with less than 3 years of observations (Kaneohe, Iceland, BOBOA, M2, SEAK, SOFS, BTM, TAO165E, TAO155W, NH-10, and JKEO). Years shown are the earliest dates of seawater $pCO_2$ trend emergence for each time series, which is the ToE estimate plus the time series start year (Table 1). These years of trend emergence and associated uncertainty are also shown in Table 2. For the pH data sets with long enough time series to calculate ToE (i.e., the circles in Fig. 2), there is no significant difference between ToE of $pCO_2$ and pH.**




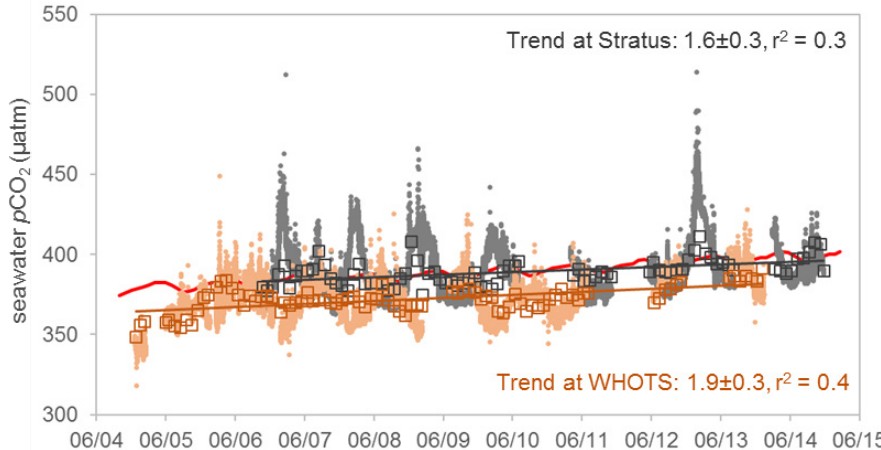

**Figure 6: Surface seawater $p$CO$_2$ (µatm) 3-hourly observations (dots), deseasoned monthly anomalies (squares), and trends (lines) for the Stratus (gray) and WHOTS (orange) time series. The time series in red is monthly averaged atmospheric $x$CO$_2$ (µmol mol$^{-1}$) from Mauna Loa, Hawaii (NOAA ESRL Global Monitoring Division, 2016). Dates are MM/YY.**