# Peer review of "Autonomous seawater *p*CO2 and pH time series from 40 surface buoys and the emergence of anthropogenic trends"

_Earth System Science Data, 2018_

## Referee Comment (RC1) · Anonymous Referee #1 · 12 Nov 2018

**Review of**
**Autonomous seawater pCO$_2$ and pH time series from 40 surface buoys and the emergence of anthropogenic trends**
by Sutton et al.

November 2018

Recommendation: Acceptable for publication after minor revisions

**1 Summary**

Sutton et al. release a comprehensive data product for $p$CO$_2$ and pH (among other variables) from 40 surface ocean buoys around the globe. Further, this paper briefly analyzes the time series data to compute Time of Emergence (ToE) of the anthropogenic emissions signal. They propose conservative estimates of ToE, since their relatively short time series do no capture the influence of decadal variability. The data product is extremely accessible and the website is well put together. One can acquire plots of near real-time pH and $p$CO$_2$ via the web server as well as select a buoy of interest from a map to retrieve well-labeled and quality-controlled data. I suggest that this manuscript

be published in $ESSD$ following minor revisions. I only have a few very minor comments/clarifications.

**2 Major Comments**

1. I appreciate the attention to detail on limitations to ToE with such short time series (*i.e.*, taking an estimate of decadal variability on the TAO buoys and applying that to all other stations). However, I'd be curious to see what the influence of the differing IAV estimates does to the ToE estimate. *I.e.*, what is the difference in the ToE when using the detrended vs. not detrended estimate of anomalies in Equation 1? I imagine that the 12% change in IAV from this tactic might propagate a decent bit of uncertainty into ToE (that is separate from the decadal variability uncertainty).

**3 Minor Comments**

2. Lines 31–33 (pg. 3): "...magnification of the seasonal amplitude of $pCO_2$ due to warming, ...resulting in increased detection time." You could cite Kwiatkowski and Orr (2018) and Landschützer et al. (2018) here, which cover this topic.

3. Lines 1–3 (pg. 6): Perhaps expand here on what future efforts will be done to improve IAV estimates. What can be done other than waiting for longer time series to develop?

4. Lines 10–11 (pg. 9): "Since ToE is dependent on the variability ...tend to have longer ToE estimates." I would suggest more clear wording for this

sentence. In the case of this application, ToE is mainly variability-induced, since all stations share a commonly imposed trend of $2\mu atm\ yr^{-1}$. However, in many cases, long ToE estimates can be also driven by a weak signal, and short ToE estimates by a very strong signal, etc.

5. Figures 1 and 2: When using a discrete color bar, it is generally advised that the tick marks align with discrete color boundaries. In their current format, both color bars have tick marks placed arbitrarily within color bounds, which makes these color divisions useless. E.g., in Figure 1, setting 10 color boundaries with colorbrewer would align the ticks/color boundaries in $25\mu$atm increments.

6. Figure 3: I suggest changing the color scheme for (b) and (c) to be mindful of those that are red-green color blind.

**References**

Lester Kwiatkowski and James C. Orr. Diverging seasonal extremes for ocean acidification during the twenty-first century. *Nature Climate Change*, 8(2):141–145, 2018. doi: 10.1038/ s41558-017-0054-0.

Peter Landschützer, Nicolas Gruber, Dorothee C. E. Bakker, Irene Stemmler, and Katharina D. Six. Strengthening seasonal marine CO2 variations due to increasing atmospheric CO2. *Nature Climate Change*, January 2018. ISSN 1758-678X, 1758-6798. doi: 10.1038/s41558-017-0057-x.

---

## Referee Comment (RC2) · Anonymous Referee #2 · 10 Dec 2018

Major comment Dr. Sutton and colleagues presented a readily accessible data product of autonomous pCO2 and pH time series from 40 surface buoys from 2004 in open ocean, coastal and coral reef sites, that exhibit extensive daily and interannual variability. Using a time of trend emergence methodology, they estimated the length of time for an anthropogenic trends in oceanic pCO2 and pH to emerge from natural variability in the 40 time series. Only at two time series datasets (WHOTS and Stratus), surface oceanic pCO2 significantly increased. However, pH time series data are too short to estimate long-term anthropogenic trends. In addition, description of pH sensor isn't detailed, compared from pCO2 sensor [Sutton et al., 2014b]. I cannot confirm post-calibrated and quality-controlled pH data (at NCEI data archive) through comparison

with in-situ calibration, discrete samples and so on, because pH sensor performance was often limited by biofouling [Bresnahan Jr et al., 2014]. After revising the manuscript to address this comment and the specific comments below, I would support publication of the author's submission.

Minor comments Figure 1 I think that only locations and names of 40 fixed moored time series station map is convenient for readers.

Line 22, Page 7 How long is it necessary for pH time series to determine a robust estimate of IAV?

---

## Referee Comment (RC3) · Anonymous Referee #3 · 15 Dec 2018

In this manuscript, the authors present a data package that incorporates measurements from 40 buoys with pCO2 and, in some cases, also pH sensors. The authors make a good case for why this dataset is of additional value compared to getting data independently from each buoy at NCEI. The authors also provide time of trend emergence estimates where the record is long enough and compare results for open ocean, coastal, and coral reef sites. This makes the paper interesting not just for potential users of the data, but also for researchers that might want to compare their own data trends to data from these buoys.

I appreciated the specific section on data availability and how to use and properly

acknowledge the dataset, which apparently is still too complicated for some data users.

This manuscript and product are timely and will be very useful for a variety of researchers, so I recommend publication after addressing the following minor issues:

Page 4 lines 10-15: what type of equilibrator is used? Is it a membrane? Page 4, line 20-26: At what temperature is pHT reported? Is there enough data at this point to evaluate the most adequate of the two sensors for long term monitoring? Page 9, lines 26-28. How likely do you think it is that this warm event will happen again? If you are discussing ToE and this event could happen again in the next 1-2 decades, wouldn't it make sense to keep it in the record for the ToE calculations and comparisons?

Minor edits: Page 2 Line 30: change "although" for "however" Page 4, line 20: add reference to Table 1 Page 8, lines 22-23: "reflecting the influence of short term of the local active reef community" please rewrite this.

---

## Author Comment (AC1) · 15 Feb 2019

We thank all referees for their thoughtful and constructive comments and suggestions on our manuscript "Autonomous seawater pCO2 and pH time series from 40 surface buoys and the emergence of anthropogenic trends." The revised manuscript will be much improved as a result of the careful critiques. Below we discuss the comments from Referee #1 point by point including original referee comments and our responses bulleted (–) underneath.

Sutton et al. release a comprehensive data product for pCO2 and pH (among other variables) from 40 surface ocean buoys around the globe. Further, this paper briefly

analyzes the time series data to compute Time of Emergence (ToE) of the anthropogenic emissions signal. They propose conservative estimates of ToE, since their relatively short time series do no capture the influence of decadal variability. The data product is extremely accessible and the website is well put together. One can acquire plots of near real-time pH and pCO2 via the web server as well as select a buoy of interest from a map to retrieve well-labeled and quality-controlled data. I suggest that this manuscript be published in ESSD following minor revisions. I only have a few very minor comments/clarifications.

2 Major Comments

1. I appreciate the attention to detail on limitations to ToE with such short time series (i.e., taking an estimate of decadal variability on the TAO buoys and applying that to all other stations). However, I'd be curious to see what the influence of the differing IAV estimates does to the ToE estimate. I.e., what is the difference in the ToE when using the detrended vs. not detrended estimate of anomalies in Equation 1? I imagine that the 12% change in IAV from this tactic might propagate a decent bit of uncertainty into ToE (that is separate from the decadal variability uncertainty).

– We find that ToE estimates are on average 55% shorter using detrended monthly anomalies compared to ToE estimated using not detrended monthly anomalies (page 9 line 21). This is different from the detrending applied to the WHOTS example for IAV. IAV is typically calculated on data with the long-term trend removed; however, we did not do this, as the long-term trend is unknown at most sites and the time series are relatively short (<12 years). The 12% change was presented here to highlight the uncertainty in the IAV estimates, which are separate from the ToE calculation.

3 Minor Comments

2. Lines 31–33 (pg. 3): ". . . magnification of the seasonal amplitude of pCO2 due to warming, . . . resulting in increased detection time." You could cite Kwiatkowski and Orr (2018) and Landschützer et al. (2018) here, which cover this topic.

– Good suggestion. Those references have been added.

3. Lines 1–3 (pg. 6): Perhaps expand here on what future efforts will be done to improve IAV estimates. What can be done other than waiting for longer time series to develop?

– Good point. We have added the following to that section: "Future efforts to improve these IAV estimates can rely on future assessment of longer time series (moored or observations from other platforms) and regional models that better characterize all modes of temporal variability."

4. Lines 10–11 (pg. 9): "Since ToE is dependent on the variability . . . tend to have longer ToE estimates." I would suggest more clear wording for this sentence. In the case of this application, ToE is mainly variability-induced, since all stations share a commonly imposed trend of 2 $\mu$atm yr-1. However, in many cases, long ToE estimates can be also driven by a weak signal, and short ToE estimates by a very strong signal, etc.

– We agree it was confusing to mention the imposed long term trend here and have modified the sentence to focus on the correlation between variability and ToE: "In this application ToE is dependent on the variability in the data, resulting in the pattern where sites that exhibit larger seasonal to interannual variability (Figs. 1 and 2) tend to have longer ToE estimates (Fig. 5)." We also suspect that our use of the term "emergence" may add confusion. Multi-ensemble modeling assessments of emergence of a forced trend over model variability typically also use the emergence terminology. This manuscript addresses a slightly different approach in assessing the time period of observations required to detect a long-term trend above natural variability. As such, throughout the manuscript we have added more description of this observation-based trend detection time approach of the method.

5. Figures 1 and 2: When using a discrete color bar, it is generally advised that the tick marks align with discrete color boundaries. In their current format, both color bars have

tick marks placed arbitrarily within color bounds, which makes these color divisions useless. E.g., in Figure 1, setting 10 color boundaries with colorbrewer would align the ticks/color boundaries in 25 $\mu$atm increments.

– Thank you for catching that. Color bar modified to align ticks/color boundaries.

6. Figure 3: I suggest changing the color scheme for (b) and (c) to be mindful of those that are red-green color blind.

– Again, thank for you catching that. Color scheme modified.

---

## Author Comment (AC2) · 15 Feb 2019

We thank all referees for their thoughtful and constructive comments and suggestions on our manuscript "Autonomous seawater pCO2 and pH time series from 40 surface buoys and the emergence of anthropogenic trends." The revised manuscript will be much improved as a result of the careful critiques. Below we discuss the comments from Referee #2 point by point including original referee comments and our responses bulleted (–) underneath.

Major comment Dr. Sutton and colleagues presented a readily accessible data product of autonomous pCO2 and pH time series from 40 surface buoys from 2004 in open

ocean, coastal and coral reef sites, that exhibit extensive daily and interannual variability. Using a time of trend emergence methodology, they estimated the length of time for an anthropogenic trends in oceanic pCO2 and pH to emerge from natural variability in the 40 time series. Only at two time series datasets (WHOTS and Stratus), surface oceanic pCO2 significantly increased. However, pH time series data are too short to estimate long-term anthropogenic trends. In addition, description of pH sensor isn't detailed, compared from pCO2 sensor [Sutton et al., 2014b]. I cannot confirm postcalibrated and quality-controlled pH data (at NCEI data archive) through comparison with in-situ calibration, discrete samples and so on, because pH sensor performance was often limited by biofouling [Bresnahan Jr et al., 2014]. After revising the manuscript to address this comment and the specific comments below, I would support publication of the author's submission.

– We agree that thorough sensor evaluation and data quality control is critical to confirming pH data quality. Entire publications are dedicated to this topic, like Bresnahan et al. 2014 cited by the reviewer. Similar to the pCO2 sensor evaluation of Sutton et al. 2014b, the 2016 paper describes in detail the moored pH sensor evaluation and data quality control, which are primarily through comparison to discrete data and independently calculated pH. That analysis determined these sensors (once calibrated and adjusted in the case of the SeaFET) have a total uncertainty of <0.02 in this particular surface buoy application. We agree with the reviewer that this point needed clarification, and we've added the following statement to that section: "Data quality control of these pH time series, including calibration, comparison with discrete samples, and assessment of drift due to sensor performance and biofouling, are described in detail by Sutton et al. (2016)."

Figure 1 I think that only locations and names of 40 fixed moored time series station map is convenient for readers.

– Very good point that we failed to link Figure 1 with the detailed site information in Table 1. We've added the following to the Figure 1 caption: "Moored time series locations

and names are detailed in Table 1." Also of note, while Figure 1 focuses on illustrating surface seawater pCO2 mean, seasonal amplitude, and IAV, the data product at NCEI (https://www.nodc.noaa.gov/ocads/oceans/Moorings/ndp097.html) includes a figure solely focused on buoy location and names for data users ease.

Line 22, Page 7 How long is it necessary for pH time series to determine a robust estimate of IAV?

– In this manuscript, we are using the pCO2 estimate of 3 years of continuous measurements (page 19 line 4; page 21 line 7) as the cutoff for presenting IAV, and as of the assessment described in this manuscript, no pH time series meet that length. Included in the IAV methodology section (page 5 line 34 – page 6 line 4) is a discussion of the uncertainty in these IAV estimates.

---

## Author Comment (AC3) · 15 Feb 2019

We thank all referees for their thoughtful and constructive comments and suggestions on our manuscript "Autonomous seawater pCO2 and pH time series from 40 surface buoys and the emergence of anthropogenic trends." The revised manuscript will be much improved as a result of the careful critiques. Below we discuss the comments from Referee #3 point by point including original referee comments and our responses bulleted (−) underneath.

In this manuscript, the authors present a data package that incorporates measurements from 40 buoys with pCO2 and, in some cases, also pH sensors. The authors

make a good case for why this dataset is of additional value compared to getting data independently from each buoy at NCEI. The authors also provide time of trend emergence estimates where the record is long enough and compare results for open ocean, coastal, and coral reef sites. This makes the paper interesting not just for potential users of the data, but also for researchers that might want to compare their own data trends to data from these buoys.

I appreciated the specific section on data availability and how to use and properly acknowledge the dataset, which apparently is still too complicated for some data users. This manuscript and product are timely and will be very useful for a variety of researchers, so I recommend publication after addressing the following minor issues:

Page 4 lines 10-15: what type of equilibrator is used? Is it a membrane?

– This is a bubble-type equilibrator. The MAPCO2 methodology is described in detail in Sutton et al. 2014b. We have added these details to the following sentence in the referenced section: "Seawater $xCO_2$ equilibration occurs by cycling a closed loop of air through an floating bubble equilibrator at the sea surface for 10 minutes, which is described in detail by Sutton et al. (2014b)."

Page 4, line 20-26: At what temperature is pHT reported? Is there enough data at this point to evaluate the most adequate of the two sensors for long term monitoring?

– We have added to line 24 that pHT is reported at in situ SST. Evaluating the two sensors requires both an analysis of existing data as presented here and targeted side-by-side test deployments of both sensors at select mooring time series sites. Because of the latter requirement, we believe this evaluation is outside of the scope of this manuscript.

Page 9, lines 26-28. How likely do you think it is that this warm event will happen again? If you are discussing ToE and this event could happen again in the next 1-2 decades, wouldn't it make sense to keep it in the record for the ToE calculations and

comparisons?

– To our knowledge, there have not been any assessments predicting future likelihood of similar North Pacific warm anomalies; however, we do cite Bond et al. 2015, which proposes the mechanisms that influenced development of the 2014-2015 anomaly. We do indeed include the 2014-2015 data in the ToE calculation for WHOTS. The section referenced by the reviewer is on the separate calculation of trends. We remove the anomalous event because it occurs at the endpoint of the time series, disproportionally influencing the linear regression as described in the more detailed trend assessment of Sutton et al. 2017 cited in this section.

Page 2 Line 30: change "although" for "however"

– Done.

Page 4, line 20: add reference to Table 1

– Good suggestion. Done.

Page 8, lines 22-23: "reflecting the influence of short term of the local active reef community" please rewrite this.

– Thank you for pointing that out. Rewritten as: "reflecting the influence of short-term (∼1-2 days) carbonate chemistry variability of the local active reef community"